# Simulation on Heat Transfer and Emergency Protection of Tanks in a Tank Farm under Fire Scenario

**DOI:** 10.3390/ijerph20075348

**Published:** 2023-03-31

**Authors:** Yingwei Bao, Feng Zhang, Jiaji Cheng, Yapeng Wang, Yu Guan, Junjie Ren, Fangbo Jin, Yunfei Cheng, Weilun Xie

**Affiliations:** College of Environment and Safety Engineering, Qingdao University of Science and Technology, 53 Zhengzhou Road, Qingdao 266042, China

**Keywords:** storage tank, heat transfer, emergency, simulation, fire

## Abstract

It is very important to understand the heat transfer process between storage tanks in a tank farm under a fire scenario, which is one of the key factors in determining the consequences of accident development. In this paper, a CFD simulation is used to study the heat transfer process and emergency protection of tanks under a fire scenario. The simulated results show that the changes in wind speed can affect the heat transfer of the tank farm. The highest temperature of the tanks at 5.3 m/s (wind speed) is 1432 K, while the highest temperature at 17.1 m/s (wind speed) is 1556 K. At the same time, the changes in wind direction can also affect the heat transfer of the tank farm. For the 45° east by north (wind direction), almost all tanks in the tank farm are affected by the fire. When the water curtain was applied as an emergency protection measure, the simulated highest temperature of the tanks decreased to 779 K (the cooling water intensity 6 L/min·m^2^), while the highest temperature of the tanks was 1432 K without water curtain protection under the actual fire conditions. The simulated highest temperature of the tanks decreased to 671 K when the emergency thermal insulation coating was sprayed on the surface of the tanks, which can effectively protect the adjacent tanks from being destroyed.

## 1. Introduction

In recent years, with the rapid development of the economy, the demand for chemical products is continuously increasing, and more and more large storage tank farms are put into use. Chemical storage tanks store flammable, explosive, toxic or harmful chemicals, which are also accompanied by fire or explosion risk in the process of transportation, storage and use [1,2]. These serious accidents may cause casualties, property losses and environmental damage. Moreover, secondary fire or explosion of adjacent tanks and even the whole tank farm may also occur due to the domino effect, resulting in more serious consequences [3]. Usually, two main approaches are used to study the heat transfer process between tanks, i.e., experimental analysis and numerical simulations. Compared with experiment analysis, a numerical simulation based on fluid dynamics has the characteristics of low cost, easy operation and relatively reliable results. Therefore, simulation research has been paid more and more attention by researchers [4,5]. 

A large number of simulation research has been carried out, from a single storage tank fire to a tank farm fire. Many parameters of storage tank fires, such as wind speed, burning rate, tank material and filling rate, are changed to investigate their effect on the consequence of the fire. Scarponi et al. [6,7] studied the impact of fire on domestic liquefied petroleum gas storage tanks. A method was proposed to evaluate the vulnerability of liquefied petroleum gas storage tanks, and the effectiveness of this method was verified with the CFD simulation tool, which provided reliable data for the safe distance of domestic liquefied petroleum gas storage tanks. At the same time, the temperature distribution of the LPG tanks partially exposed to a pool fire were simulated, which could affect the liquid flow inside the tank and cause the internal pressurization of the whole tank to increase. Yet-Pole et al. [8] used a CFD tool to simulate a storage tank explosion accident in a petrochemical plant. The instantaneous distribution of dangerous substances at different locations and the serious consequences were obtained, which provided an effective basis to explain the cause of the accident and reduce the impact of the accident. Jiang et al. [9,10] designed a fire experiment of a storage tank and considered the factors affecting the thermal response of the storage tank, such as fire type and tank type, etc. Accordingly, a more accurate model was proposed when combined with the CFD numerical simulation tool. However, till now, the heat transfer process of a tank farm under fire scenario is still not fully understood due to the multi-factor complex interactions. Moreover, the simulation of emergency protective measures in a tank farm after accidents are discussed far less.

Thermal protective measures need to be considered sufficiently in order to reduce the impact of the accident on adjacent tanks. Traditionally, there are mainly four kinds of thermal protective measures for adjacent storage tanks in fire, namely, safe distances between tanks, fire dike, water spray protection and coating protection [11]. The safe distances between tanks are determined in the design stage of the tank farm, which is generally related to the capacity of the firefighting water tank in the tank farm. Ghasemi et al. [12] proposed a new parameter on the basis of the amount of cooling water and the safe distances between tanks. Accordingly, at least 25% of the tank’s area was saved, which can still ensure the tanks will not be damaged by thermal radiation. However, in the case of a large fire accident, these four traditional thermal protective measures cannot ensure that the requirements of protecting adjacent storage tanks will be met completely [13]. 

Recently, new emergency thermal insulation coatings for the thermal protection of storage tanks were designed and prepared successfully in our laboratory [14,15]. These new coatings are expected to be used as an accident emergency product, which will be sprayed onto the walls of adjacent storage tanks in the case of a fire. In this way, the adjacent tanks will be protected due to the thermal insulation performance of the emergency products. 

In this paper, a fire accident in a chemical tank farm is selected, and the CFD simulation tool is used to study the heat transfer process of the tank farm under fire. The factors such as wind speed and wind directions that affect the heat transfer of the tank farm are also analyzed. Moreover, the emergency measures, such as water curtain protection and emergency thermal insulation coating, were considered and simulated to investigate their effects on the heat transfer of the tank farm.

## 2. The Description of the Fire Accident

There are 21 storage tanks (three rows from north to south) in the accidental tank farm. In the first row, there are five 2000 m^3^ storage tanks labeled with 1# to 5# from east to west, which store about 9100 tons of materials. The second row contains eight 1000 m^3^ storage tanks, all of which are empty. In the third row, there are eight 1000 m^3^ storage tanks, among which tank 6# in the southeast corner stores about 889 tons of materials. The layout of the tank farm is shown in Figure 1.

On 31 May 2021, a fire accident took place in tank 1# of the tank farm in Hebei, China, which affected many adjacent tanks in the tank farm. The fire lasted for 84 h. Through accident investigation, the experts determined that the materials in the tanks mainly contained methane, ethane and other hydrocarbons.

The cause of the accident was illegal hot work and failure to install a flame arrester and shut-off valve in the oil vapor recovery line, which ignited the combustible gas in the tank. The accident site is shown in Figure 2 and Figure 3. In the early stage of the accident, firefighters focused on extinguishing and cooling tanks 1# and 2#–5#. The tanks on the south side were not protected and were also in danger due to thermal radiation.

The temperature of the day was 290 K, the main wind direction was northeast, 71° east by north, wind level three, about 3.4–5.4 m/s, and there was no rainfall process.

## 3. Establishment of the Model

### 3.1. Governing Equation

The fluid flow of the tank farm fire belongs to turbulent flow and follows conservation equations of mass, momentum and energy, which can be expressed by Equations (1)–(3) [16]:

Mass conservation equation:(1)$$\frac{\partial \rho_{f}}{\partial t}+\nabla \cdot (\rho_{f}\cdot \vec{v})=0$$
where $\rho_{f}$ is the fluid density, and  v → is the speed.

Momentum conservation equation:(2)$$\frac{\partial }{\partial t}\left(\rho_{f}\cdot \vec{v}\right)+\nabla \cdot \left(\rho_{f}\cdot \vec{v}\cdot \vec{v}\right)=-\nabla p+\nabla \left(\overset{=}{\tau }\right)+\rho_{f}\cdot g$$
where p is static pressure, *τ* is the stress tensor and g is the gravitational constant.

Energy conservation equation (ignoring the heat loss of the tank to the environment):(3)$$\frac{\partial }{\partial t}\left(\rho E\right)+\nabla \cdot \left(\vec{\nu }\left(\rho E+p\right)\right)=\nabla \cdot \left(k_{\mathrm{eff}}\cdot \nabla T+\left(\overset{=}{\tau }\cdot \vec{\nu }\right)\right)$$
where E is energy, k_eff_ is the effective thermal conductivity and T is temperature.

A realizable k-*ε* model is selected for simulating tank farm fires. The key parameter equations in the model are expressed by Equations (4) and (5) [17]:

Turbulence energy equation k: (4)$$\rho \frac{\mathrm{dk}}{\mathrm{dt}}=\frac{\partial }{\partial \chi_{i}}\left[\left(\mu +\frac{\mu_{t}}{\delta_{k}}\right)\frac{\partial k}{\partial \chi_{i}}\right]+G_{k}+G_{b}-\rho \varepsilon -Y_{M}$$

Dissipation rate equation *ε*: (5)$$\rho \frac{d\varepsilon }{\mathrm{dt}}=\frac{\partial }{\partial \chi_{i}}\left[\left(\mu +\frac{\mu_{t}}{\delta_{\varepsilon }}\right)\frac{\partial \varepsilon }{\partial \chi_{i}}\right]+C_{1\varepsilon }\frac{\varepsilon }{k}\left(G_{k}+C_{3\varepsilon }G_{b}\right)-C_{2\varepsilon }\rho \frac{\varepsilon^{2}}{k}$$
where G_k_ is the turbulent energy generation term due to the mean velocity gradient; G_b_ is the turbulent energy generation term generated by a buoyancy effect, which is 0 in this paper; Y_M_ is the effect of compressible turbulent pulsation expansion on the total dissipation rate, and the value in this paper is 0; C_1*ε*_, C_2*ε*_ and C_3*ε*_ are empirical constants, the default values of which are 1.44, 1.92 and 0.09, respectively; ∂k is the Prandtl number corresponding to turbulence energy k, the default value of which is 1.0; *∂ε* is the Trumpian number corresponding to the dissipation rate *ε*, the default value of which is 1.3.

The combustion gas is assumed to be an ideal gas and shown with state Equation (6): (6)pV=nRT
where *R* is the mole gas constant in J/(mol·K).

### 3.2. Calculation Domain and Geometric Model

The fluid region and size of the model need to be determined. The fluid region is too small to accurately determine simulation results due to computational non-convergence. Too large a fluid region will lead to increased time costs and consume more computing resources [18]. When the fluid at the outlet of the fluid domain has been sufficiently developed, the model simulation results are slightly different from the actual situation, and the fluid region can be determined [19]. To sum up, the three-dimensional model region determined in this paper is 120 × 75 × 15 m. A total of 21 tanks in the tank farm are included in the model, among which 2000 m^3^ tanks are φ16 m × 10 m, and 1000 m^3^ tanks are φ11.6 m × 9.5 m. The wall thickness was set as 10 mm for both types of tanks.

The model of the accident tank farm is shown in Figure 4, in which tank 1# is calculated as the entrance due to a fire, and the rest of the surrounding boundaries are calculated as the exit.

Considering the influence of calculation accuracy and calculation time, unstructured grids are adopted for division, and the total number of grids is 822,000, as shown in Figure 5.

In this paper, five different grid numbers were established for the independent grid study [20], and the number of cells was 371 thousand, 518 thousand, 619 thousand, 822 thousand and 1.025 million, respectively. The initial attributes of each parameter were as follows: the pressure was set as standard atmospheric pressure; temperature 290 K; the ambient air inlet and fuel inlet were 4 m/s. The ambient air inlet consists of nitrogen 0.79 and oxygen 0.21. For the sake of simplification, the fuel inlet was assumed to be methane with a ratio of 1. The outlet was set as the pressure outlet; the remaining boundaries were set to the walls. The temperature at the center point of tank 15# was monitored when the simulated accident occurred in 120 s, and the results are shown in Figure 6. The results show that the temperature does not change significantly as the number of meshes increases and gradually stabilizes. At the same time, an increase in the number of grids results in an increase in computational time. Accordingly, 822 thousand grids were chosen as the grid-independent solution for this simulation, taking into account both accuracy and cost.

### 3.3. Solution Method

#### 3.3.1. Condition Assumption

Due to the fact that the tank farm fire is affected by various factors and the process is complicated and uncertain, it is difficult to guarantee complete agreement with the actual situation in the simulation process. On the basis of comprehensive consideration of workstation performance and simulation accuracy, the following assumptions are made: ① it is assumed that only tank 1# caught fire and burned, while the other tanks did not; ② it is assumed that the tanks in the tank farm are not deformed in the fire, and only the temperature changes of the tanks are considered.

#### 3.3.2. Boundary Conditions

The ambient air inlet was set as velocity-inlet. For the temperature setting, the ambient temperature on the day of the accident (290 K) was selected. The pressure was set to the standard atmosphere; the wind speed refers to the level three wind speed (3.4–5.4 m/s) on the day of the accident, which was 5.3 m/s. The wind direction was 71° east by north. The composition of the air was 0.79 nitrogen and 0.21 oxygen.

The fuel inlet was set as velocity-inlet: the temperature was 290 K. The wind speed was set as 5.3 m/s, and the direction was 71° east by north. The fuel option was methane, with a fuel ratio of 1 at the fuel inlet.

The pressure outlet was set as the gas outlet, and K and Epsilon were chosen for the turbulence gauge method.

The ground and remaining walls were set to be non-slip wall boundaries, and the temperature was taken to be 290 K (ambient temperature).

After checking the mesh, the gravity option was turned on, and the gravity size was set to 9.8 m/s^2^; the energy equation was opened; the realizable k-ε model was adopted for the turbulence model. The DO radiation model and component transport model were opened. The non-premixed combustion model was selected, and the PDF (probability density function) table was generated. Coupling and pseudo-transient methods were used to accelerate convergence, followed by initialization and iterative computation.

#### 3.3.3. Determination of the Simulation Results

The convergence criteria for the residual errors of the continuity, momentum, volume fraction and turbulence equations were set to 1 × 10^−3^, and for the energy equation, they were set to 1 × 10^−6^. After iterative calculations, the residual curves satisfied the convergence criteria, and the computational convergence was determined. In the following simulations, the results of jet-fire burning in 1800 s are discussed.

## 4. Results and Discussion

### 4.1. Simulation Results of the Fire Accident That Occurred in the Tank Farm

The temperature distribution of the tanks in the tank farm was analyzed using the CFD-Post 2020R1 software, as shown in Figure 7. The tanks 1#, 6#–14#, which were located downwind of the accident tank, were most affected by the fire. The highest temperature of the tanks could reach 1432 K. Under this extremely high temperature, an explosion would probably occur, and secondary accidents would result in more serious consequences in the tank farm. Tanks 15#–21# were least affected by the thermal radiation of the accident tank. Compared with tanks 6#–14#, the temperatures of tanks 15#–21# were lower, which made it easier to not cause serious consequences, such as rupture or failure. The combustion temperature of hydrocarbon combustion reached 2060 K, which appeared at the south exit, as shown in Figure 8. Therefore, the potentially catastrophic consequences need to be paid more attention due to the severely high local temperatures.

The velocity vector of the tank farm air–fluid domain is shown in Figure 9. It can be seen that the wind speed on the empty fire road was higher, and the tank wall’s temperature near the fire road was also higher. At the same time, the gas flow speed across the gap between the tanks such as 1#, 6#–14# will also increase. The tank wall’s temperature in these areas was also higher than that of the other tanks.

### 4.2. Effect of Wind Speed on Heat Transfer of the Tank Farm

Wind speed is an important factor affecting fire development. According to the local meteorological record where the accident occurred, the maximum wind speed level is 7–8, i.e., about 13.9 to 20.7 m/s. Here, a wind speed of 17.1 m/s was selected as an input parameter of the simulation model for the tank farm fire. Additional simulation parameters are consistent with those in Section 3.3.2. The temperature results of tanks in the tank farm are shown in Figure 10, and the temperature results of tanks 2#–5# are shown in Figure 11.

Compared with Figure 7 (the wind speed is set as 5.3 m/s), it can be seen from Figure 10 (the wind speed is 17.1 m/s) that the main affected area was storage tanks 6#–12#. The highest temperature of the tank can reach 1556 K, which was 124 K higher than that when the wind speed was 5.3 m/s. The mass percentage composition of substances at the outlet was analyzed through the post-processing software. The results indicated that when the wind speed increased, the mass fraction of CO_2_ at the outlet increased from 13.19% to 14.08%, the mass fraction of CO increased from 13.79% to 13.90%, and the mass fraction of CH_4_ decreased from 22.93% to 16.46%. This proves that the increased wind speed helps to improve combustion efficiency, and more heat would be produced. Li et al. [21] proposed the correlation Equation (7) of flame inclination angle based on hydrocarbon fuel combustion experiment data combined with flame propagation. It can be seen from the equation that the flame inclination is proportional to the wind speed. When the wind speed increases, the angle of the flame also increases, causing the downwind distance to increase, resulting in the downwind tank being closer to the flame and receiving more thermal radiation.
(7)$$\tan(\theta )\propto \frac{u^{2}}{gl}$$
where *θ* is the flame inclination angle; *u* is wind speed, m/s; *g* is the acceleration of gravity, m/s^2^; *l* is the width in the horizontal direction of the flame.

Accordingly, the downwind tanks 6#–12# were affected more seriously, whereas the affected area was significantly reduced. It can be seen from Figure 11 that only the temperature of tanks 2#–5# can reach above 500 K, and the temperatures of the remaining tanks were all below 460 K. As mentioned above, when the wind speed increases, the number of affected tanks will decrease. However, the higher temperature of the tanks may cause a domino effect, causing secondary disasters and increasing the range of affected areas. Therefore, for the tank farm fire rescue, in addition to firefighting operations, attention should be paid to the protection of the surrounding environment and the protection of firefighters themselves.

### 4.3. Effect of Wind Direction on Heat Transfer of the Tank Farm

Wind direction plays a crucial role in the spread of the fire. The fire occurred in tank 1#, which was located northeast of the tank farm. Therefore, east, north and northeast winds can affect the tank farm. According to the local meteorological record in 2021, a total of 346 days of weather conditions were recorded. The statistical results are shown in Figure 12, and 151 days of wind directions are east, north and northeast, respectively.

In addition to the simulated wind direction (71° east by north) in Section 4.1, four other wind directions, east wind, 19° east by north wind, 45° east by north wind and north wind, were selected, respectively, as the simulated wind direction. The temperature distribution of tanks in the tank farm was simulated under these four different wind directions. In the setting of simulation parameters, only the wind direction is changed, and the other parameters are consistent with those in Section 3.3.2. Accordingly, the simulated results are shown in Figure 13, Figure 14, Figure 15, Figure 16, Figure 17, Figure 18, Figure 19 and Figure 20.

As shown in Figure 13, the impact of the east wind on the tank farm was mainly concentrated in tanks 2#–5#, and the highest temperature of the tanks reached 1419 K. It was easy to cause a secondary fire or explosion accidents. Therefore, the fire must be put out quickly, and special protection measures should be used for tanks 2#–5#. As shown in Figure 14, due to the isolation of the fire road and the change in the wind direction (east), the combustible gas did not arrive at tanks 6#–21# directly. At the same time, the entrained air with ambient temperature from the wind had a cooling effect; thus, the temperature of tanks 6#–21# is relatively lower, which was not susceptible to the fire.

The 19° east by north wind (Figure 15) and the simulated wind direction in Section 4.1 had a similar influence on the tank farm. The difference was that the affected tanks were different. For the two wind directions, tanks 13# and 14# were both affected, with temperatures exceeding 870 K. As shown in Figure 16, the temperatures of tanks 13#–21# were relatively high, which was in the downward direction of the wind. Accordingly, sufficient oxygen was provided to make the fuel burn more sufficiently. At the same time, these tanks (13#–21#) prevented the combustion gas from moving towards the southwest direction, forcing the gas to move towards the north, resulting in the temperature increase in tanks 4# and 5#.

As shown in Figure 17, the 45° east by north wind had the greatest impact on the tank farm. Almost all tanks in the tank farm were affected, and the temperatures of most tanks were above 800 K. These high temperatures might result in a domino effect on the whole tank farm and deteriorate the impact of the fire accident. As shown in Figure 18, this wind can transport the combustible gas through tanks 6#–21#, which can result in the temperature rise of these tanks due to the heat release of the fire. At the same time, some of the combustion gas was obstructed by these tanks and moved towards tanks 2#–5#, which resulted in the temperature rise of tanks 2#–5#.

The north wind had the lowest effect on the tank farm, as shown in Figure 19 and Figure 20. Only tanks 6#–10# were highly affected, and the local highest temperature was close to 1300 K. Special attention should be paid to tank 6# due to its higher temperature, which may result in secondary fire or explosion. 

As per previous analysis, the 45° east by north wind had the greatest influence on the tank farm, and its influence scope covers almost the whole tank farm. The impact of the east wind on the tank farm was also serious because the affected tanks 2#–5# contained combustible material. Once the tank’s temperature is too high, leakage or explosion may occur, and the consequences are destructive.

### 4.4. Effect of Water Curtain Protection on the Tank Farm

In the early stage of the accident, firefighters only used firefighting water to cool down tanks 1#–5# on the north side and did not provide timely and effective protection measures on tanks 6#–21# on the south side. As a result, tank 6# was ignited due to the too-high temperature, causing the deterioration of the accident. In order to reduce the impact of the tank fire on the adjacent tanks and prevent the occurrence of a domino effect in the tank farm, it was necessary to carry out cooling operations or water curtain protection measures to protect these adjacent tanks. Here, based on the original model, a spray inlet was established to simulate the effect of water curtain protection on the southern tanks. According to the requirements of the amount of cooling water in relevant literature [22,23,24], the flow rate of cooling water was set as the minimum of 2 L/min·m^2^ and the maximum of 6 L/min·m^2^ to study the cooling effect of water spraying with different intensity on storage tanks.

The water spraying entrance was set on the right side of the fire road. The water spray inlet set at this position can not only form a water curtain to effectively protect the southern tank but also have a certain blocking effect on the combustion gas released from tank 1#. The location of the water spray inlet is shown in Figure 21.

Liquid water was selected as the input material parameter, and the temperature was consistent with the ambient temperature (290 K). The intensity of the cooling water was set as 2 L/min·m^2^. The other parameter settings are consistent with those in Section 3.3.2. The simulation results from Section 4.1 are compared to analyze the emergency protection effect of the water curtain.

As shown in Figure 22, the simulation results showed that the water curtain had an obvious effect on tanks 7#, 9# and 11#, and the temperature was reduced by about 140 K. It can be seen from Figure 23 that the water curtain can form an obvious protective layer to block the combustible gas released from tank 1# and protect the row of tanks near the fire road. However, in the southmost row of tanks, combustible gas that was not fully extinguished by the water curtain could continue to burn, resulting in a temperature rise in these tanks. At this time, the highest temperature appeared in tanks 8# and 10#, which was about 1377 K. Compared with the results from Section 4.1, the highest temperature was decreased by 55 K, which proved the effectiveness of the water curtain.

In order to obtain a better cooling effect of the water curtain, only the intensity of the incident cooling water was increased to 6 L/min·m^2^, and other input parameters were not changed. The simulation results are shown in Figure 24.

As shown in Figure 24, when the cooling water intensity increased to 6 L/min·m^2^, the temperature of the tank farm was significantly decreased. Most of the tanks’ temperature was below 620 K, and the highest temperature was 779 K in tanks 7# and 9#, which is 653 K lower than that without water curtain protection. When the cooling water intensity increased from 2 to 6 L/min·m^2^, the highest temperature in the tank farm decreased from 1377 to 779 K, and the cooling protection effect was very obvious.

### 4.5. Effect of Emergency Thermal Insulation Coating on the Tank Farm

Water curtain emergency protection has a good cooling effect. However, its protective performance is affected by various factors. For example, when the intensity of cooling water increases, a lot of cooling water will be consumed. Firefighting water storage capacity is limited and sometimes cannot completely meet the requirement of firefighting. To overcome the shortcomings of current emergency protection measures, new intumescent insulation emergency coatings for thermal protection of storage tanks are developed in our laboratory [14,15]. In the case of fire, the emergency thermal insulation coating is sprayed on the surface of the tank to prevent heat transfer.

In this paper, the method provided by Xie et al. [15] was selected to prepare emergency thermal insulation coating PPMHI-L (potassium polyacrylate and organically modified hectorite and intumescent flame retardant—Lotus root starch). Firstly, hectorite was fully dispersed in deionized water to obtain inorganic gel solutions. The gel solution was transferred to a three-necked flask and activated for 30 min at 333 K. Then, the gel solution was prepared by anhydrous ethanol, KH-570, and deionized water in the ratio of 72:20:18. When the gel solution was dried, the organically modified hectorite was obtained. Organically modified hectorite, sodium bisulfite, potassium acrylate and ammonium persulfate were mixed and polymerized to prepare PPMH material (potassium polyacrylate and organically modified hectorite). Lotus root powder, ammonium polyphosphate and melamine were selected as an intumescent flame-retardant system and added to PPMH material. Accordingly, thermal insulation material PPMHI-L was prepared. The specific formula is shown in Table 1.

The emergency thermal insulation coating is defined in the simulation software and named PPMHI-L. In the established model, a new layer was added to the outside of the tank wall, and its material parameter was selected as PPMHI-L. In order to ensure the simulation accuracy, the thickness of the coating model is defined as 3 cm according to the final thickness of the expansion layer in the experiment. The coating model is shown in Figure 25. The original model has a grid number of 1.03 million.

Under N_2_ atmosphere (flow rate of 60 mL/min), samples of about 10 mg were put into an aluminum crucible and heated from 303 to 973 K at a heating rate of 20 K/min. The thermogravimetric (TG) analyzer was used to dynamically detect the pyrolysis process of the thermal insulation materials. According to the thermogravimetric (TG) analysis results of PPMHI-L (Figure 26), the coating decomposes water vapor at about 373 K and loses about 38% of the total mass of the material. Therefore, in terms of simulation parameters, the content of gaseous water in the air is increased. According to the calculation of the tank data, the required coating area is about 10,700 m^2^, and these coatings can decompose about 3% water vapor in the total space of the model. Therefore, the gas composition of air and ambient air inlet is changed to nitrogen 0.7663, oxygen 0.2037, and water vapor 0.03. Since PPMHI-L decomposes water vapor at about 373 K, for simplicity, the initial temperature of the simulation was set at 373 K. Additional parameters are the same as those in Section 3.3.2.

As shown in Figure 27, the highest temperature of the tank farm with PPMHI-L thermal insulation coating was only 671 K, which appeared in tanks 7# and 9#. Compared with the simulation result without PPMHI-L thermal insulation coating (Figure 7), the highest temperature of the whole tank farm was decreased by 761 K, showing a highly obvious thermal insulation effect. 

## 5. Conclusions

A fire accident in a storage tank farm was selected for simulation. The heat transfer process of the tank farm under fire was simulated. At the same time, the effect of wind speeds and wind directions on the heat transfer of the tank farm was analyzed. Finally, the effect of emergency measures, such as water curtain protection and emergency thermal insulation coating, was also simulated. The main conclusions are as follows: (1)The highest simulated temperature 1432 K occurred in tanks 6#–14#, downwind of the accident tank. The remaining tanks were rarely affected by the flames and were mainly heated up by thermal radiation;(2)When the wind speed increases, the affected area of the tank farm decreases compared with the low wind speed. However, the highest temperature of the tanks increased significantly, especially tanks 6#–14#, and the highest temperature that could be reached was 1556 K;(3)Among the selected wind directions (east wind, 71° east by north, 19° east by north wind, 45° east by north wind and north wind), 45° east by north wind has the greatest influence on the tank farm, and its influence scope covers almost the whole tank farm;(4)Compared with the cooling water intensity of 2 L/min·m^2^, when the cooling water intensity increases to 6 L/min·m^2^, the temperature of the tank farm is significantly decreased. The highest temperature of the tank is 779 K, which is 653 K lower than the result without water curtain protection;(5)PPMHI-L emergency thermal insulation coating has a great thermal insulation effect. The simulated highest temperature of the tanks decreases to 671 K when the emergency thermal insulation coating PPMHI-L is sprayed on the surface of the tanks, which can effectively protect the adjacent tanks.

## Figures and Tables

**Figure 1 ijerph-20-05348-f001:**
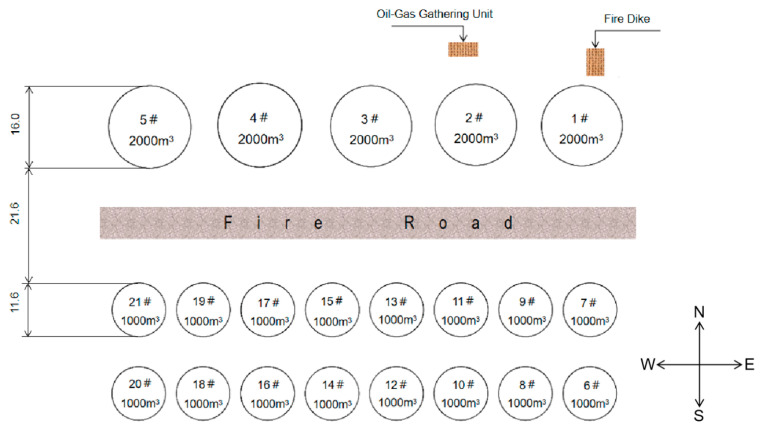
The layout of the tank farm.

**Figure 2 ijerph-20-05348-f002:**
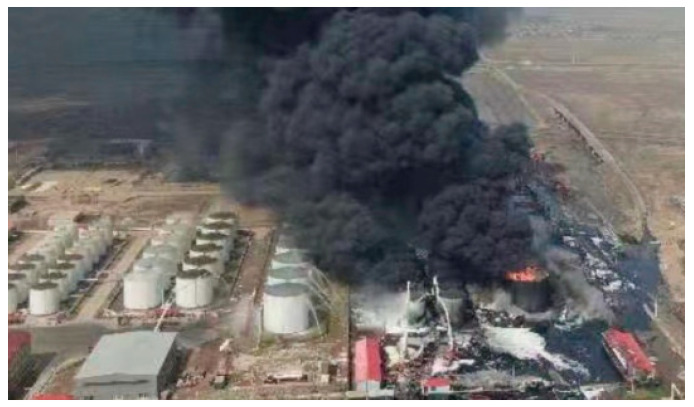
Tank farm fire accident.

**Figure 3 ijerph-20-05348-f003:**
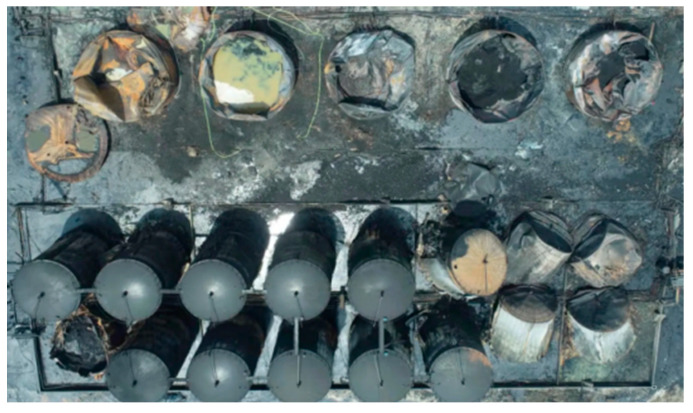
Tank farm after the accident.

**Figure 4 ijerph-20-05348-f004:**
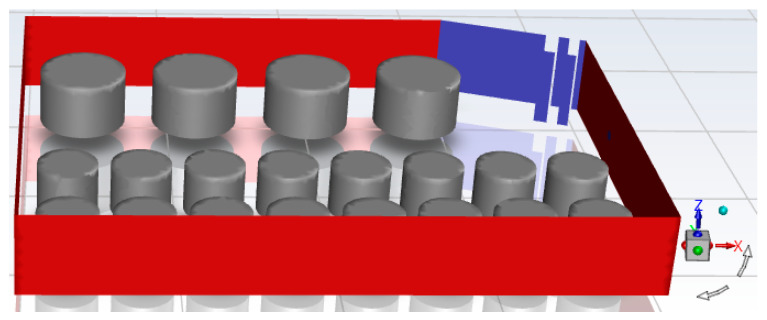
Model diagram of accidental tank farm.

**Figure 5 ijerph-20-05348-f005:**
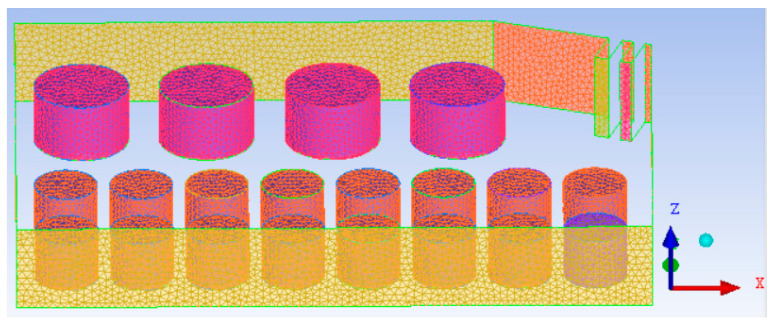
The mesh of the computational domain.

**Figure 6 ijerph-20-05348-f006:**
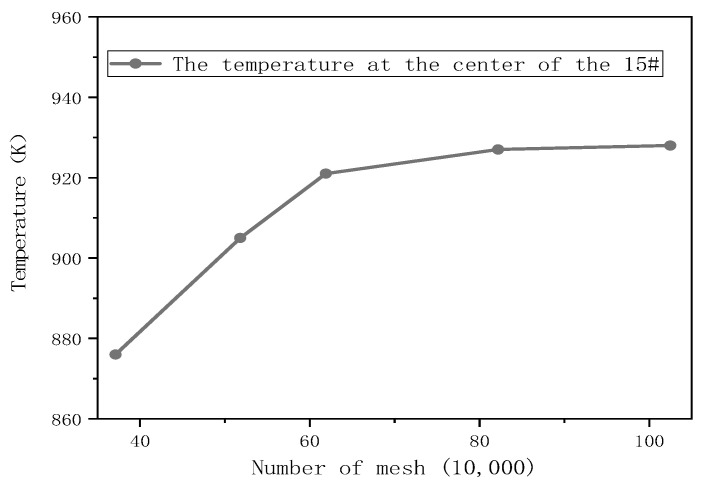
Center point temperature diagram of 15# tank at 120 s under five meshes.

**Figure 7 ijerph-20-05348-f007:**
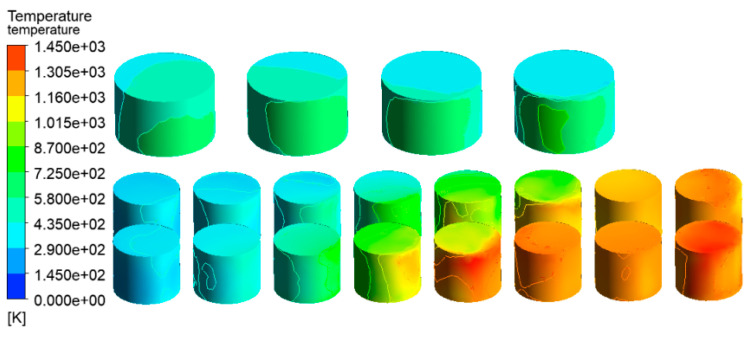
Temperature diagram of tanks in the tank area (V = 5.3 m/s).

**Figure 8 ijerph-20-05348-f008:**
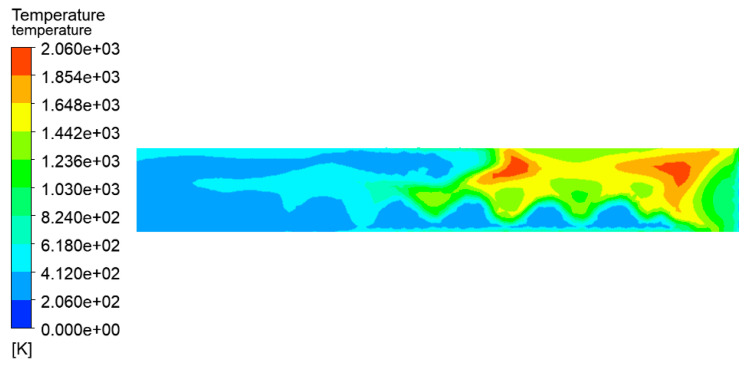
Temperature diagram of the south exit section (V = 5.3 m/s).

**Figure 9 ijerph-20-05348-f009:**
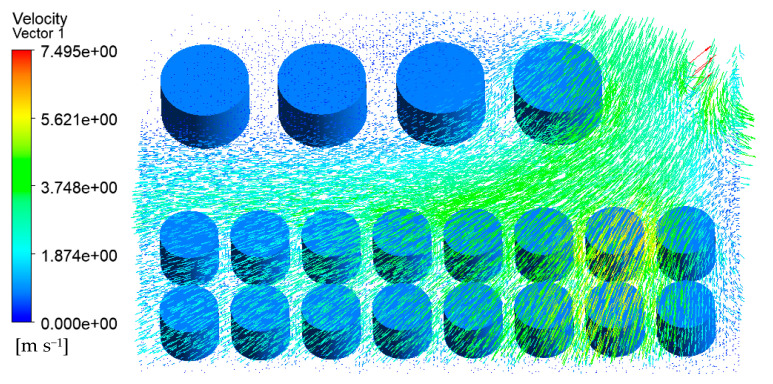
Velocity vector diagram of the air–fluid domain in the tank area (V = 5.3 m/s).

**Figure 10 ijerph-20-05348-f010:**
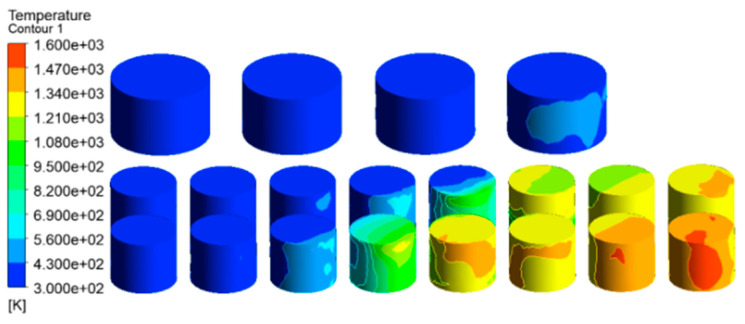
Temperature diagram of tanks in the tank area (V = 17.1 m/s).

**Figure 11 ijerph-20-05348-f011:**
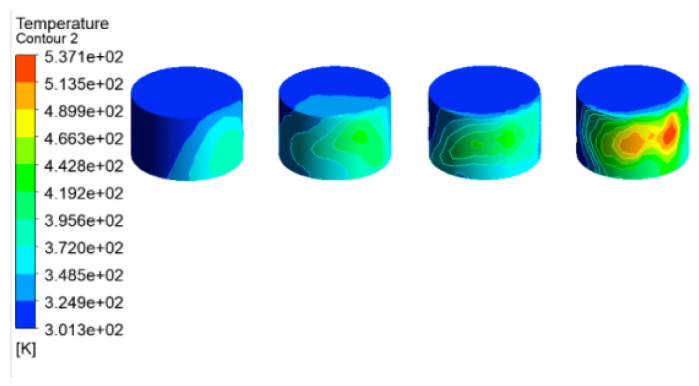
Temperature diagram of tanks 2#–5# (V = 17.1 m/s).

**Figure 12 ijerph-20-05348-f012:**
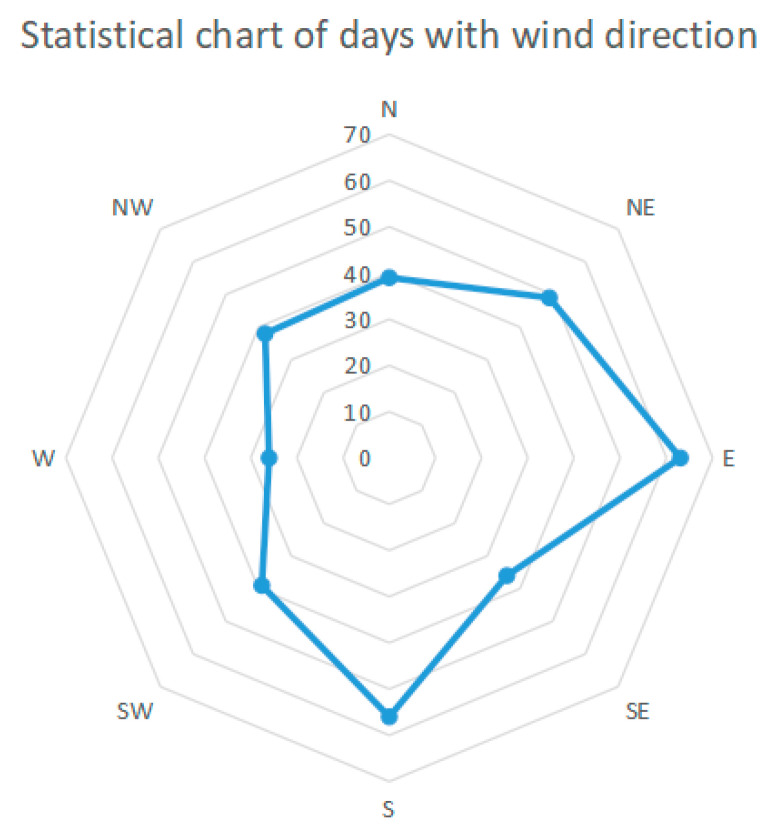
Statistical chart of days with wind direction.

**Figure 13 ijerph-20-05348-f013:**
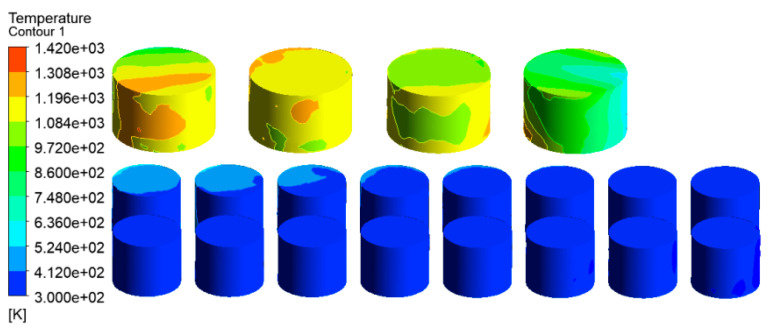
Temperature diagram of the tank area in the east wind.

**Figure 14 ijerph-20-05348-f014:**
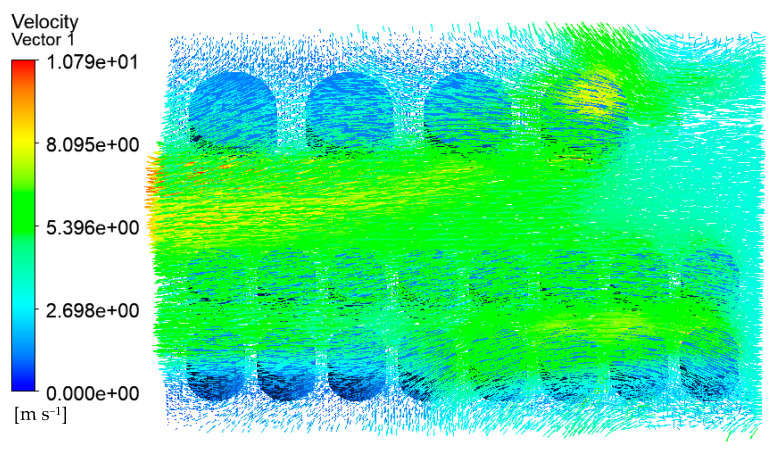
Velocity vector diagram of the air–fluid domain in the east wind tank area.

**Figure 15 ijerph-20-05348-f015:**
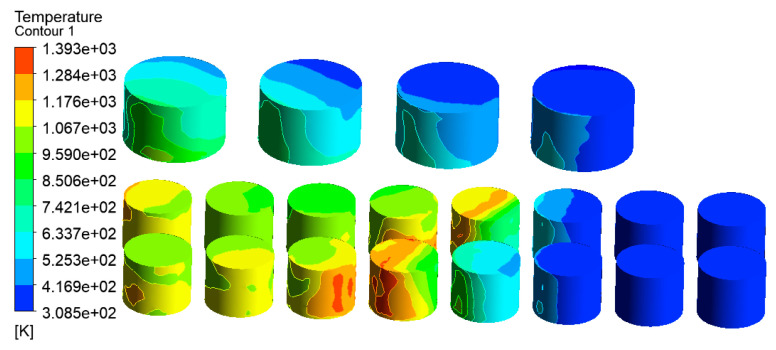
Temperature diagram of tank farm in the 19° east by north air tank farm.

**Figure 16 ijerph-20-05348-f016:**
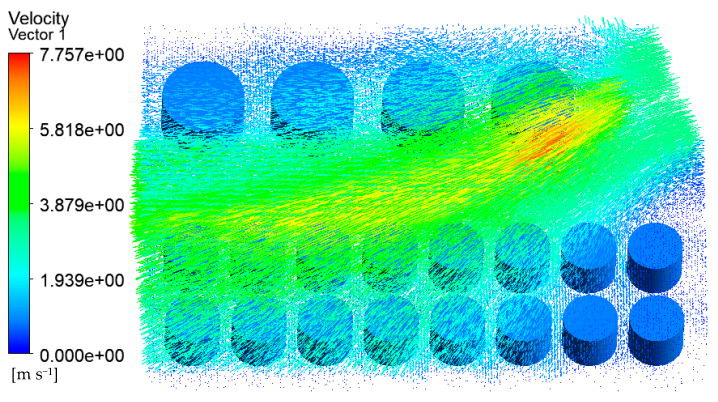
Velocity vector diagram of the air–fluid domain in the 19° east by north air tank.

**Figure 17 ijerph-20-05348-f017:**
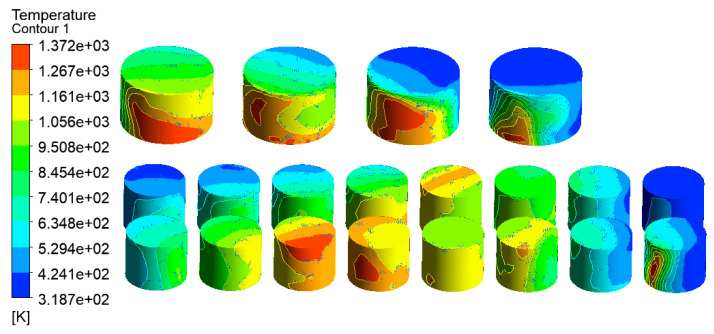
Temperature diagram of the tank farm in the 45° east by north air tank farm.

**Figure 18 ijerph-20-05348-f018:**
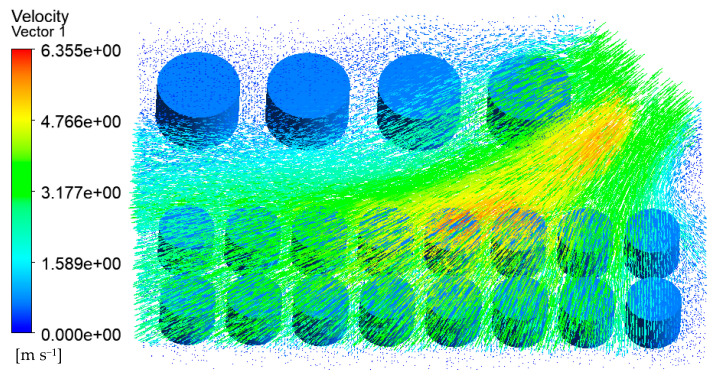
Velocity vector diagram of the air–fluid domain in the 45° east by north wind tank.

**Figure 19 ijerph-20-05348-f019:**
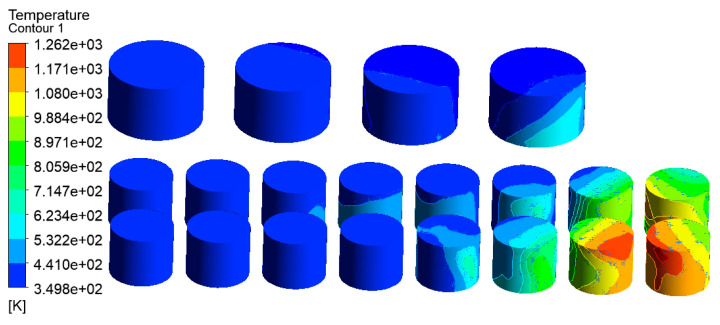
Temperature diagram of the tank area in the north wind.

**Figure 20 ijerph-20-05348-f020:**
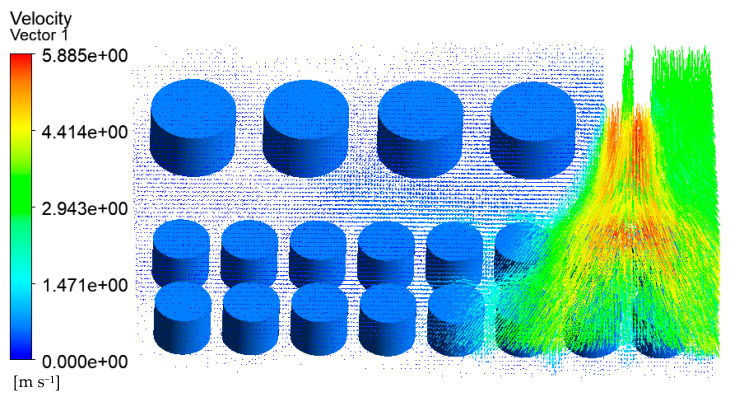
Velocity vector diagram of the air–fluid domain in the north wind tank area.

**Figure 21 ijerph-20-05348-f021:**
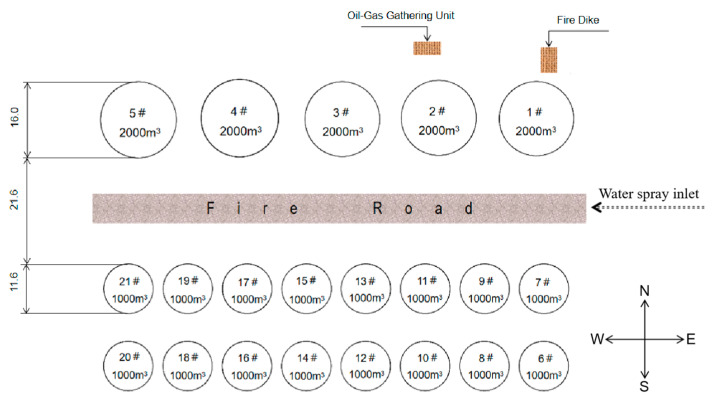
Layout of the sprinkler inlet.

**Figure 22 ijerph-20-05348-f022:**
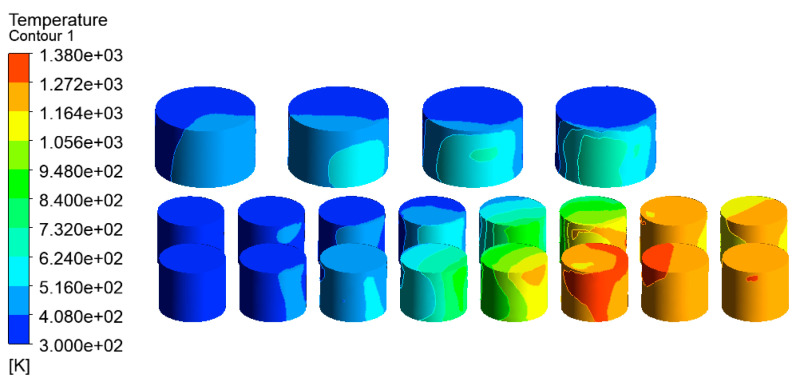
Water curtain emergency protection simulation temperature diagram (Vwater = 2 L/min·m^2^).

**Figure 23 ijerph-20-05348-f023:**
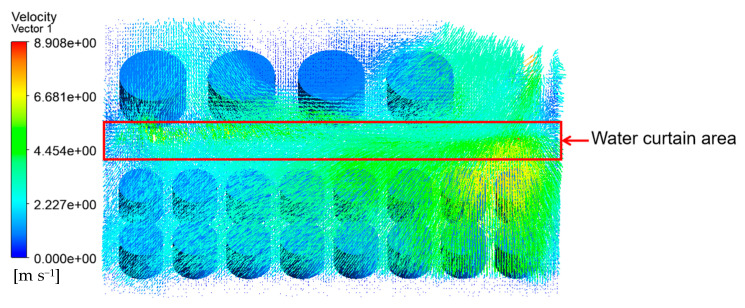
Flow field velocity vector diagram of water curtain emergency protection simulation (Vwater = 2 L/min·m^2^).

**Figure 24 ijerph-20-05348-f024:**
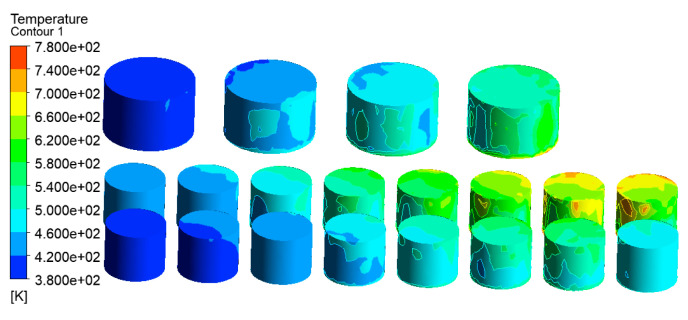
Water curtain emergency protection simulation temperature diagram (Vwater = 6 L/min·m^2^).

**Figure 25 ijerph-20-05348-f025:**
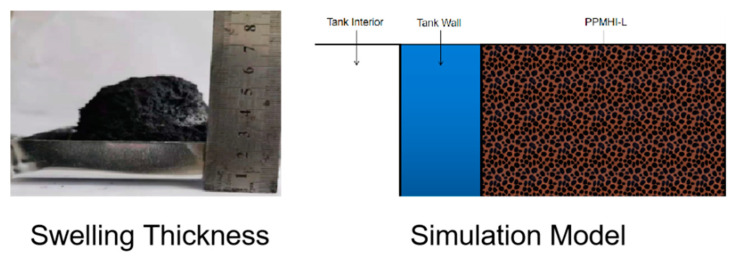
Expansion thickness and coating model.

**Figure 26 ijerph-20-05348-f026:**
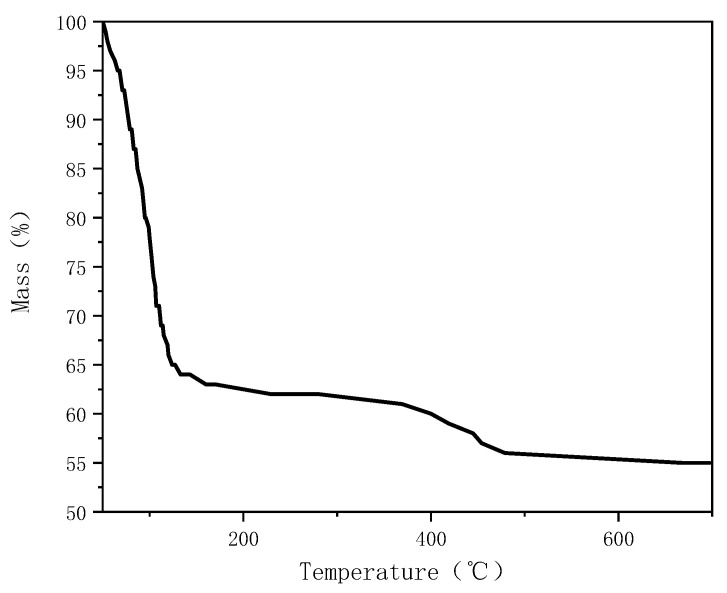
TG of PPMHI-L.

**Figure 27 ijerph-20-05348-f027:**
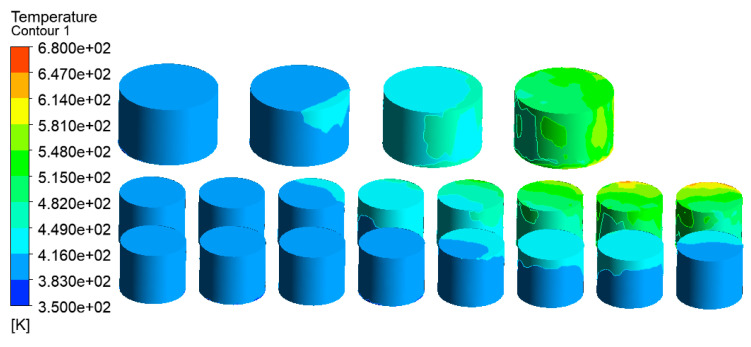
Simulation results of PPMHI-L thermal insulation coating emergency protection.

**Table 1 ijerph-20-05348-t001:** PPMHI-L formulation.

Materials	PPMH/%	APP/%	Melamine/%	Lotus Root Starch/%	H_2_O/%
PPMHI-L	31.92	0.43	0.11	0.26	67.26

## Data Availability

Not applicable.

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
