# Peer review of "Simulation on Heat Transfer and Emergency Protection of Tanks in a Tank Farm under Fire Scenario"

_ijerph, 2023, doi:10.3390/ijerph20075348_

Round 1

Reviewer 1 Report

In this paper, the authors state that they have discussed the heat transfer processes occurred in multiple tank fires using numerical analysis. However, although the paper shows the temperature distributions on the surface of the tanks, there is no discussion of the heat transfer processes. Therefore, the discussion in this paper is not sufficient. In particular, there is no verification of the calculation results obtained in this paper. As a result, we have to say the reliability of the paper's results is extremely low. Thus, this paper does not include valuable examination, discussion, and information as a scientific and technical paper. Based on the above, this reviewer thinks that this paper should be rejected for publication in this journal.

Author Response

Reviewer #1: In this paper, the authors state that they have discussed the heat transfer processes occurred in multiple tank fires using numerical analysis. However, although the paper shows the temperature distributions on the surface of the tanks, there is no discussion of the heat transfer processes. Therefore, the discussion in this paper is not sufficient. In particular, there is no verification of the calculation results obtained in this paper. As a result, we have to say the reliability of the paper's results is extremely low. Thus, this paper does not include valuable examination, discussion, and information as a scientific and technical paper. Based on the above, this reviewer thinks that this paper should be rejected for publication in this journal.

Response: Thank you very much for the reviewer’s comments. In this paper, we analyze the temperature profile of the tank during fire and the effect of different factors on the heat transfer in the tank farm. We did not perform additional experimental validation due to the potential effect of the tanks fire on the environment. At the same time, the cost of the tank farm fire is very high. In this paper, five different grid numbers were established for grid independence study. The results show that the temperature does not change significantly as the number of meshes increases and gradually stabilizes, which can ensure the accuracy of the simulation results.

Reviewer 2 Report

In this paper, numerical simulation method was used to study the heat transfer of tanks under fire conditions, and the emergency protection measure was proposed to protect adjacent tanks, which has practical significance for fire emergency protection of tank area. My comments are listed as follows

1) The description of preparation of PPMHI-L thermal insulation material is not sufficient and more details should be added.

2) The thermogravimetric (TG) analysis was inadequate and the conditions under which the test was performed were not specified clearly.

3) Some of the figures are not clear enough, for example, Figures 1 and 21 in the article.

Author Response

Please see the attachment, thank you!

Reviewer 3 Report

(1) Page 8, line 235: We know that increasing the wind speed will bring more fresh air and also take more heat away. Therefore, it is suggested to estimate the oxygen required for combustion to further reveal the reasons for different temperature rises. Is it possible that the different angle of inclination of flame leads to different radiation angle, which also leads to different temperature rise.

(2) Conclusion: It is suggested to write the conclusion in sections, which will be more conducive to reading.

(3) Please use the tense correctly and use the past tense more frequently. For example, the results observed in this paper should be expressed in the past tense. The past tense should also be used in the Results and Discussion, Summary, and Conclusion.

Author Response

Please see the attachment, thank you!

Round 2

Reviewer 1 Report

The authors’ claim that "the cost of the tank farm fire is very high." in Response is completely correct.

However, in order to examine large-scale heat and mass transport phenomena from the results of small-scale heat and mass transport experiments, scaling laws, such as Nusselt, Reynolds, Froude, and Grashof numbers and so on, have been developed in the science of heat and mass transfer. A lot of the results obtained in the heat and mass transfer research are fundamentally organized by their dimensionless numbers and the validity of the results is verified.

Therefore, by using scaling law, it is possible to check whether the numerical results obtained by the authors are realistic and reasonable. In fact, equation (7), which the authors presented in their revised manuscript, is a Froude number.

Nevertheless, the fact that the validity of the numerical results for fluid dynamics and heat transport phenomena has not been verified at all is a critical problem of this paper.

Based on the above considerations, it is not worthy to be published as this paper in IJERPH.
